# Homothorax controls a binary Rhodopsin switch in *Drosophila* ocelli

**Abhishek Kumar Mishra**[1]*, **Cornelia Fritsch**[1], **Roumen Voutev**[2], **Richard S. Mann**[2], **Simon G. Sprecher**[1]*

**1** Institute of Cell and Developmental Biology, Department of Biology, University of Fribourg, Fribourg, Switzerland, **2** Department of Biochemistry and Molecular Biophysics and Neuroscience, Mortimer B. Zukerman Mind Brain Behavior Institute, Columbia University, New York, United States of America

* abhimishra313@gmail.com (AKM); simon.sprecher@gmail.com (SGS)

**Data Availability Statement:** All relevant data are within the manuscript and its Supporting Information files.

**Funding:** This work is supported by the Swiss National Science foundation grant number

## Abstract

Visual perception of the environment is mediated by specialized photoreceptor (PR) neurons of the eye. Each PR expresses photosensitive opsins, which are activated by a particular wavelength of light. In most insects, the visual system comprises a pair of compound eyes that are mainly associated with motion, color or polarized light detection, and a triplet of ocelli that are thought to be critical during flight to detect horizon and movements. It is widely believed that the evolutionary diversification of compound eye and ocelli in insects occurred from an ancestral visual organ around 500 million years ago. Concurrently, opsin genes were also duplicated to provide distinct spectral sensitivities to different PRs of compound eye and ocelli. In the fruit fly *Drosophila melanogaster*, Rhodopsin1 (Rh1) and Rh2 are closely related opsins that originated from the duplication of a single ancestral gene. However, in the visual organs, Rh2 is uniquely expressed in ocelli whereas Rh1 is uniquely expressed in outer PRs of the compound eye. It is currently unknown how this differential expression of Rh1 and Rh2 in the two visual organs is controlled to provide unique spectral sensitivities to ocelli and compound eyes. Here, we show that Homothorax (Hth) is expressed in ocelli and confers proper *rhodopsin* expression. We find that Hth controls a binary Rhodopsin switch in ocelli to promote Rh2 expression and repress Rh1 expression. Genetic and molecular analysis of *rh1* and *rh2* supports that Hth acts through their promoters to regulate Rhodopsin expression in the ocelli. Finally, we also show that when ectopically expressed in the retina, *hth* is sufficient to induce Rh2 expression only at the outer PRs in a cell autonomous manner. We therefore propose that the diversification of *rhodpsins* in the ocelli and retinal outer PRs occurred by duplication of an ancestral gene, which is under the control of Homothorax.

## Author summary

Sensory perception of light is mediated by specialized photoreceptor neurons of the eye. Each photoreceptor expresses unique photopigments called opsins and they are sensitive to particular wavelengths of light. In insects, ocelli and compound eyes are the main

310030_188471 to SGS, the Novartis foundation for biomedical research grant number 18A017 to SGS and the National Institute of Health grant number 5R35GM118336-05 to RSM. The funders had no role in study design, data collection and analysis, decision to publish, or preparation of the manuscript.

**Competing interests:** The authors have declared that no competing interests exist.

photosensory organs and they express different opsins. It is believed that *opsins* were duplicated during evolution to provide specificity to ocelli and the compound eye and this is corelated with their distinct functions. We show that Homothorax acts to control a binary Rhodopsin switch in the fruit fly *Drosophila melanogaster* to promote Rhodopsin 2 expression and represses Rhodopsin 1 expression in the ocelli. Genetic and molecular analysis showed that Homothorax acts through the promoters of *rhosopsin 1* and *rhosopsin 2* and controls their expression in the ocelli. We also show that Hth binding sites in the promoter region of *rhodopsin 1* and *rhodopsin 2* are conserved between different *Drosophila* species. We therefore proposed that Hth may have acted as a critical determinant during evolution which was required to provide specificity to the ocelli and compound eye by regulating a binary Rhodopsin switch in the ocelli.

## Introduction

The ability to perceive and discriminate a broad range of environmental stimuli in nature is essential for many aspects of life. Animals rely heavily on visual cues to perform complex tasks such as navigation to find food, mates and shelter as well as social interactions. Visual cues are perceived by visual organs that contain photoreceptors (PRs) as light-sensing structures. PRs are specialized cells that gather information from the surrounding world which is subsequently processed by the brain. Each PR expresses a unique photosensitive *opsin/rhodopsin* that defines the wavelength of light by which a PR will be activated.

It is believed that eyes have evolved separately due to fundamental differences between visual organs of different animals [1]. However, it is also known that eye development in different animal phyla shares a common genetic network initiated by *Pax6* gene orthologs [2]. Similarities in the gene regulatory network that controls eye development, further strengthen the idea that phylogenetically diverse eye types may share a conserved eye developmental program [3–5]. Insects are among the largest and most diverse animal groups. Therefore, decoding eye development in insects offer great opportunity to unravel developmental insights that lead to the emergence of evolutionary complexity.

In most insects, compound eyes represent the prominent visual organs that are responsible for providing major share of the visual information. In the fruit fly *Drosophila melanogaster*, each compound eye consists of approximately 850 ommatidia and each ommatidium houses eight PRs: six outer PRs and two inner PRs [6]. Additionally, winged insects (such as *Drosophila*) possess ocelli that are comparatively simple photosensory organs embedded in the dorsal head cuticle [7]. In *Drosophila*, a triplet of ocelli (one medial and two lateral) is arranged in a triangular shape between the two compound eyes and the dorsal vertex of the head [8]. It is believed that insect compound eye- and ocellus-like precursor structures have segregated from an ancestral eye over 500 million years ago [9,10]. The evolution of new opsin genes by gene duplication enabled these visual organs to perform different functions that require distinct spectral sensitivities [11]. In *Drosophila*, phylogenetic analysis supports that Rhodopsin1 (Rh1), Rh2 and Rh6 have originated from a common ancestral gene [12]. A first gene duplication may have separated Rh6 from Rh1/Rh2 and a second gene duplication may have separated the closely related Rh1 and Rh2 [12]. The green-sensitive Rh6 is expressed in the inner PRs of the compound eye and is critically involved in color perception [13–15]. The blue-green sensitive Rh1 is expressed in the outer PRs of the compound eye and is mainly associated with motion detection [16–18]. Conversely, the violet-sensitive Rh2 is expressed in PRs of the ocelli [19–21] and is proposed to be involved in horizon sensing and flight stabilization [22,23].

While it is known that *rh1*, *rh2* and *rh6* are genetically linked [24], it is still unknown how they are differentially expressed in different PRs in *Drosophila*.

Here we show that the homeodomain transcription factor Homothorax (Hth) regulates Rh2 expression in the ocelli. We demonstrate that Hth is expressed in ocellar PRs and controls a binary *rhodopsin* switch by promoting Rh2 expression and repressing Rh1 expression in ocelli. We also demonstrate that misexpression of Hth forces outer PR of the retina to induce Rh2 expression and clonal expression in the retina suggest that this process is cell autonomous. Furthermore, genetic and molecular analysis of *rh1* and *rh2* shows that the *rhodopsin* switch in ocelli is transcriptionally controlled by Hth and that it may act directly through *rh1* and *rh2* promoter sequences. Finally, we argue that while Hth maintains Rh3 fate in the DRA [25], it initiates Rh2 fate in the ocelli. The results presented here greatly adds to our understanding of how genetically linked opsins are spatiotemporally controlled to provide distinct spectral sensitivity to different visual organs.

## Results

### Differential expression of *rhodopsins* in the compound eye and ocelli

It is believed that the hexapod ancestor of extant insects only had a single visual organ (also referred as ancestral eye) and that other visual organs (such as compound eye and ocelli) evolved as a result of a morphological bifurcation event over 500 million years ago [10,11] (Fig 1A–1C). It is hypothesised that different functions of compound eye and ocelli in conjunction with gene duplication events of opsins led to the emergence of an ocelli specific *opsin* gene [11]. In most insects, opsins are categorised based on their distinct spectral sensitivity [26–28]. In the *Drosophila* compound eye, brightness detection is achieved by inputs from six outer PRs (R1-R6) that express the blue-sensitive Rhodopsin 1 (Rh1). *Drosophila* uses outer PRs mainly for motion detection and dim light vision [16–18]. Color vision requires two inner PRs that encode either one of the UV-sensitive Rh3 or Rh4 in R7 PRs, and either the blue-sensitive Rh5 or the green-sensitive Rh6 in R8 PRs [29,30]. In most insects, ocelli express an opsin, which is different from those present in the compound eye [11,21,31]. In *Drosophila*, PRs of the ocelli express the violet-sensitive Rh2 (Fig 1B) [19–21]. The different spectral sensitivity of ocelli is also reflected by their involvement in performing different functions than the compound eye and they are believed to detect horizon, control head orientation and stabilize flight posture while flying [22]. To get an insight into *rhodopsin* gene duplications in *Drosophila*, we compared the coding sequences of *rh1* to *rh6* by generating a phylogenetic tree (by using MUSCLE online tool; phylogeny.fr) [32]. The phylogenetic tree made by the maximum likelihood method showed two clades with branching support value of 1 each (Fig 1D). Clade I showed a tandem gene duplication that separated Rh5 from the closely related Rh3/Rh4 (Fig 1D). Clade II showed a first tandem gene duplication that separated Rh6 from Rh1/Rh2 (Fig 1D). Rh6 subsequently got expressed in the inner PRs of the compound eye (Fig 1J). A further gene duplication led to the separation of closely related Rh1 and Rh2 (Fig 1D). While Rh1 subsequently got expressed in the outer PRs of the compound eye (Fig 1H), Rh2 got exclusively expressed in the ocelli (Fig 1F). To further analyse the evolutionary origin of clade II *opsin* gene duplications, we generated a phylogenetic tree by using amino acid sequences of *Drosophila* Rh1, Rh2, Rh6 and their putative orthologs from other dipteran species (*Ceratitis capitata* or med fly, *Musca domestica* or house fly, *Glossina palpalis* or tsetse fly, *Lucilia cuprina* or Australian sheep blow fly and *Aedes aegypti/Anopheles gambiae* or mosquitoes; sequences collected from Feuda lab on bitbucket (https://bitbucket.org/Feuda-lab/opsin_diptera/src/master/) [33]. The resulting phylogenetic alignment of Rh1, Rh2 and Rh6 suggests that Rh6 is ancestral to all dipteran species including mosquitoes. A common Rh1/Rh2 ortholog

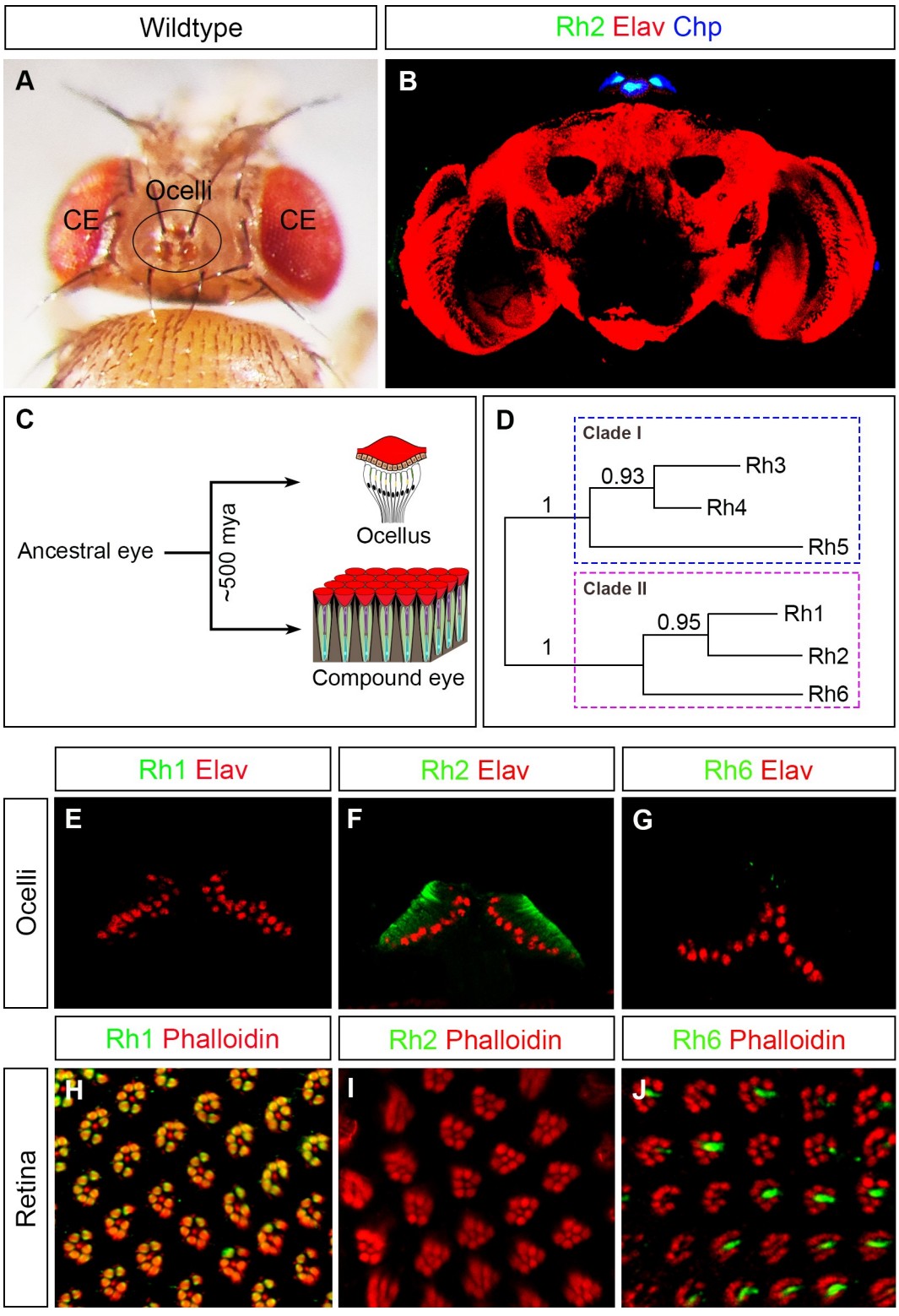

**Fig 1. Overview of an insect visual system, its evolutionary diversification and comparison of Rhodopsin expression in the *Drosophila* ocelli and compound eye.** (A) Dorsal view of the head of a fruit fly showing the positions of compound eye (CE) and ocelli (black circle; Image provided by Abhishek Kumar Mishra). (B) Cross section of a *Drosophila* adult brain where ocelli are marked by expression of Rh2 (in green). Chp expression (in blue) marked ocellar PRs and Elav expression (in red) marked neuronal population (C) Schematic representation of a morphological bifurcation event that occurred over 500

million years ago (mya) in an ancestral eye of hexapod ancestors that is believed to form ocelli and compound eye. (D) Schematic representation of gene duplications in *Drosophila rhodopsins* during evolution. Blue dotted square box marks clade I consisting of Rh3, Rh4 and Rh5 whereas violet dotted square box marks clade II consisting of Rh1, Rh2 and Rh6. A first tandem gene duplication separated Rh6 from Rh1/Rh2 and a second gene duplication separated the closely related Rh1 and Rh2. (E-G) Longitudinal sections through the ocelli stained for Rh1 (E), Rh2 (F) or Rh6 (G) shown in green. These proteins, if expressed, are located to the rhabdomeres directly below the lenses, while Elav (red counterstain), which was used to identify the position of the ocelli, localizes to the nuclei of the photoreceptors, which are positioned below the rhabdomeres (compare with schematic drawing in C). (H-J) Cross sections of photoreceptors in the retina stained for Rh1 (H), Rh2 (I) or Rh6 (J) shown in green and for Phalloidin localising to the rhabdomeres shown in red. (E, H) Rh1 is exclusively expressed in the outer PRs of the retina but not in ocelli. (F, I) Rh2 is uniquely expressed in all ocellar PRs but not in the retina. (G, J) Rh6 is normally expressed in the inner PRs in the retina but not in ocelli.

originated from a duplication of Rh6 in the lineage leading to the higher dipterans and a second gene duplication separated Rh1 from Rh2. Further gene duplications of Rh6 occurred in the mosquitoes and of Rh1 in the house fly *Musca domestica* (S1 Fig).

### Homothorax regulates a binary Rhodopsin switch in the ocelli

In the compound eye, the homeodomain transcription factor Homothorax (Hth) is expressed in the dorsal rim area (DRA) of inner PRs, where it has been shown to be both necessary and sufficient to regulate Rh3 expression and thereby critically contribute to the polarized-light sensing system [25,34]. We found that Hth is also expressed in all PRs of the ocelli (Fig 2A). Since ocellar PRs express Rh2, we first analysed if Hth is involved in regulating this expression. We performed knockdown of *hth* using the pan-photoreceptor driver *lGMR*-Gal4 and observed a complete loss of Rh2 expression in ocellar PRs (Fig 2B). Additionally, we also observed that in *hth* knockdown, the loss of Rh2 expression in the ocelli is compensated by the gain of Rh1 (Fig 2B), a Rhodopsin that is normally expressed in outer PRs of the compound eye. Thus, Hth regulates a binary switch of Rhodopsin in the ocelli where it promotes Rh2 expression and represses Rh1 expression.

Hth often acts together with Extradenticle (Exd) [35,36]. Therefore, to test if Exd is also involved in regulation of the binary Rhodopsin switch in the ocelli, we performed knockdown of *exd* with *lGMR*-Gal4 and found that it also resulted in the loss of Rh2 expression and gain of Rh1 expression in the ocellar PRs (Fig 2C).

In the compound eye, a feedback mechanism has been described that allows Rh6 to transcriptionally repress Rh5 in yellow R8 PRs [37]. Since in *hth* knockdown, ectopic Rh1 expression in the ocelli replaces Rh2, we speculated that Hth may repress Rh1 in order to allow Rh2 expression. Knockdown of *hth* in turn would remove this repression and as a result Rh1 would get activated and repress Rh2 in the ocelli. To check this hypothesis, we misexpressed Rh1 in ocelli by using *lGMR*-Gal4 and found that ectopic expression of Rh1 was not sufficient to repress Rh2 and in this case ocelli co-expressed both Rh1 and Rh2 (S2A Fig). Thus, loss of Rh2 in *hth* knockdown seems to be independent of Rh1 repression.

To further test whether switching Rhodopsin expression might also alter PR identity based on other molecular markers of outer PRs in the retina, we monitored Seven-up (svp) and BarH1 expression in the ocelli in *hth* knockdown. Svp is a steroid hormone receptor and is expressed in R3/R4 and R1/R6 pairs whereas BarH1 is a homeobox transcription factor expressed in the R1/R6 pair of the developing compound eye [38,39]. We found that neither Svp nor BarH1 are expressed in wildtype, nor in *hth* knockdown ocelli (S3A, S3B, S3C and S3D Fig). Therefore, although ocellar PRs gained Rh1 and lost Rh2 expression when *hth* was knocked down, they do not seem to undergo an identity change towards retinal outer PRs.

We next investigated whether regulation of this Rhodopsin switch by Hth occurs at the transcriptional level by using ocelli specific *rh2-lacZ* [20] and outer PR specific *rh1-lacZ* [39]

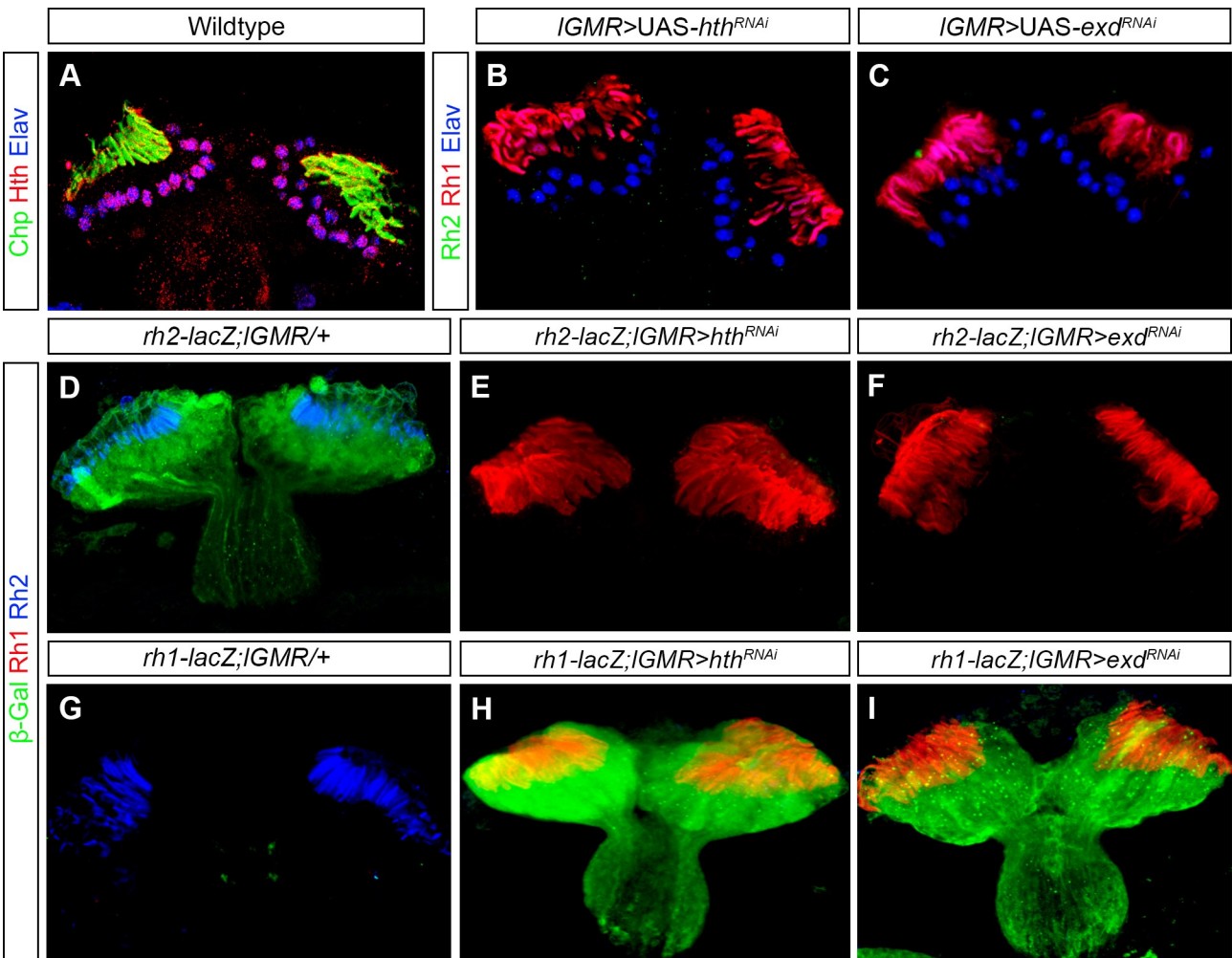

**Fig 2. Hth expression and phenotype in ocelli.** (A) Antibody staining of Hth in the wildtype ocelli showing its expression in all ocellar PRs. Chp (green) and Elav (blue) marks ocellar PR neurons. (B) *hth* knockdown (*lGMR>UAS-hth^RNAi*) in the ocelli by pan-photoreceptor *lGMR*-Gal4. In *hth* knockdown, Rh2 expression (green) is lost and there was a gain of Rh1 expression (red) in the ocellar PRs. (C) *exd* knockdown (*lGMR>UAS-exd^RNAi*) in ocelli showed a similar phenotype i.e., loss of Rh2 expression (green) and gain of Rh1 expression (red). (D, E, F) Antibody staining of β-Galactosidase (β-Gal) in the control (*rh2-lacZ; lGMR/+*), in *hth* knockdown (*rh2-lacZ; lGMR>hth^RNAi*) and in *exd* knockdown ocelli. βGal (green) expression was seen in control ocelli, also marked by Rh2 (blue) (D). However, βGal (green) expression was absent in both *hth* and *exd* knockdown ocelli, and ocellar PRs in both knockdowns were marked by Rh1 (red) (E, F). (G, H, I) Antibody staining of βGal in the control (*rh1-lacZ; lGMR/+*), in *hth* knockdown (*rh1-lacZ; lGMR>hth^RNAi*) and in *exd* knockdown ocelli. In the control, ocelli were marked by Rh2 (blue) and βGal (green) expression was absent (G) whereas βGal expression was seen ectopically in both *hth* and *exd* knockdown ocelli where ocellar PRs were marked by Rh1 (red) (H, I).

reporter lines. In wildtype control animals, *β*-gal expression was specifically observed in ocelli in case of *rh2-lacZ* (Fig 2D) whereas no expression was observed in ocelli in the case of *rh1-lacZ* (Fig 2G). In *hth* knockdown background, we found that *β*-gal expression from *rh2-lacZ* is completely abolished from the ocelli (Fig 2E), whereas ectopic *β*-gal expression from *rh1-lacZ* is now seen in ocellar PRs (Fig 2H). Therefore, Hth is indeed involved in regulating a binary Rhodopsin switch in the ocelli by promoting transcription of *rh2* and repressing transcription of *rh1*.

Since Hth and Exd act together, we next investigated whether Exd is also involved in regulating *rh1* and *rh2* transcriptionally. We indeed found that *β*-gal expression from *rh2-lacZ* was lost (Fig 2F) and ectopic *β*-gal expression from *rh1-lacZ* was observed in *exd* knockdown in ocellar PRs (Fig 2I).

We have previously shown that the homeodomain transcription factor Hazy (Flybase: Pph13 for PvuII-PstI homology 13) controls expression of Rh2 in the ocelli [40,41]. We therefore next asked if Hazy and Hth act jointly to regulate the Rhodopsin switch in ocelli. However, we find that in *hazy*[-/-] null mutant flies, absence of Rh2 expression is not accompanied with the ectopic expression of Rh1 in ocellar PRs (S2B Fig). Moreover, we found that Hth is still expressed in the ocelli of *hazy*[-/-] mutant flies (S2C Fig). Also, the expression of Hazy remained unchanged in ocellar PRs when knocking down *hth* (S2D Fig). Thus, the Hth-dependent Rhodopsin switch in the ocelli does not depend on Hazy.

## Hth controls a binary Rhodopsin switch in ocelli by acting through the promoters of *rh1* and *rh2*

Since Hth encodes a homeodomain transcription factor we next investigated if Hth may act by directly regulating the *rhodopsin* promotors. Minimal promoter sequences for *rh1* and *rh2* have been identified previously [39,20]. Like all other *Drosophila rhodopsin* promoters, they are rather short (300–400 bp) and contain a Rhodopsin-Conserved-Sequence-I (RCSI) element that provides binding sites for Pax6 orthologs and other factors that promote photoreceptor-specific expression [42]. We aligned the minimal promoter sequences of twelve different *Drosophila* species (S4 and S5 Figs) and found a high level of conservation within the twelve *rh1* promoter sequences, while the *rh2* promoters were more divergent. A direct alignment of the two promoters was not possible since they have no sequence similarities apart from the RCSI site. A portion of the *rh2* minimal promoter sequence overlaps with the coding sequence of the neighbouring gene *CG14297* (S5 Fig). We identified three potential Hth binding sites in the minimal promoter region of *rh2* (-293/+55) [20]: one within the coding sequence of *CG14297*, one directly following its Stop codon and one within its 3'UTR. While the sites in the coding sequence and in the 3'UTR are conserved, the site at the Stop codon is only present in the four closest relatives of *Drosophila melanogaster*. The *rh1* minimal promoter region (-247/+73) [39] contains two potential Hth binding sites upstream of the RCSI (S4 Fig). To test if the Hth binding sites in the *rh1* and *rh2* promoter regions may be involved in the Rhodopsin switch in ocelli, we created flies containing transgenes with the minimal promoter regions of *rh1* or *rh2* driving GFP (*rh1-GFP* and *rh2-GFP* [40]). Next, in order to abolish Hth binding, we introduced point mutations in the Hth binding regions of the *rh1* and *rh2* promoters and created *rh1(hth mut)-GFP* and *rh2(hth mut)-GFP* transgenic flies (Fig 3A–3D) (See Materials and Methods for details). In support with the previous observations, we find *rh2-GFP* expression in the ocelli (Fig 3B) whereas *rh1-GFP* is not expressed in ocellar PRs (Fig 3E). However, by mutating the Hth binding sites in the promoter region of *rh2* (*Rh2 (hth mut)-GFP*), we observed a loss of GFP expression in the ocelli (Fig 3C). Conversely, we found that deleting the Hth binding sites in the promoter region of *rh1* (*rh1 (hth mut)-GFP)* leads to ectopic GFP expression in the ocelli (Fig 3F).

## Hth is sufficient to induce Rh2 expression in the outer PRs of the retina

In the dorsal rim area of the retina, Hth is required in inner PRs to promote Rh3 expression and ectopic expression of Hth under the control of *lGMR*-Gal4 was sufficient to block the expression of inner PR Rhodopsins (Rh4, Rh5 and Rh6) and to induce Rh3 expression in all inner PRs. However, in outer PRs, expression of Rh1 was not affected suggesting that only inner PRs were responsive to Hth misexpression [25]. We next investigated if misexpression of Hth in the retina was also sufficient to induce Rh2 expression. In the wildtype retina, Rh1 was uniquely expressed in outer PRs whereas Rh2 was absent (Fig 4A, 4A' and 4A"). When Hth was misexpressed in the retina under the control of *lGMR*-Gal4, we observed that Rh2 was now ectopically

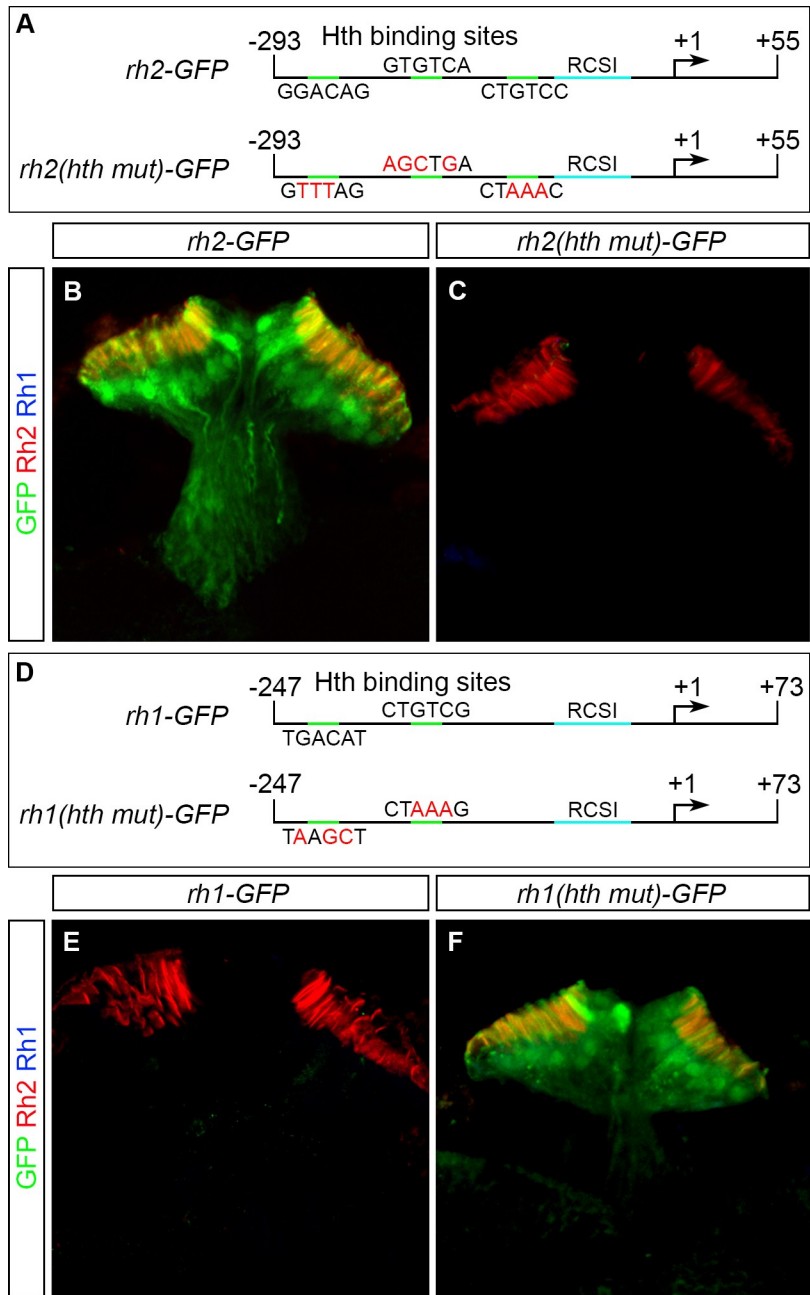

**Fig 3. Expression of *rh1* and *rh2* reporter constructs in ocelli.** (A) Schematic representation of the *rh2* minimal promoter used for making GFP reporter constructs ranging from positions -293 to +55 of the transcription start (+1, arrow). The wildtype promoter (*rh2-GFP*; in the top) contains three potential Hth binding sites (green) with their corresponding sequences shown. In the *hth* mutant promoter (*rh2(hth mut)-GFP*; in the bottom), the mutated sequences are shown with altered residues depicted in red. (B, C) Expression of the wildtype *rh2-GFP* and mutant *rh2 (hth mut)-GFP* reporter lines (green) in ocelli stained with antibodies against GFP (green), Rh2 (red) and Rh1 (blue). Wildtype *rh2-GFP* (marked by anti-GFP) is expressed in PRs of ocelli (marked by anti-Rh2) staining the entire rhabdomeres (B) whereas the mutant *rh2(hth-mut)-GFP* (marked by anti-GFP) is not expressed in ocelli (C). (D) Schematic representation of the *rh1* minimal promoter used for making the GFP reporter constructs ranging from positions -247 to +73 of the transcription start (+1, arrow). The wildtype promoter (top) contains two potential Hth binding sites (green; with its corresponding sequence shown). In the *hth* mutant promoter (bottom), the mutated sequences are shown with the altered residues depicted in red. (E, F) Expression of the wildtype *rh1-GFP* and mutant *rh1(hth mut)-GFP* reporter lines in ocelli stained with antibodies against GFP (green) Rh2 (red) and Rh1 (blue). (E) Like Rh1, the *rh1-GFP* reporter is not expressed in the PRs of ocelli expressing only Rh2. (F) The mutant *rh1(hth-mut)-GFP* reporter line is expressed in PRs of ocelli.

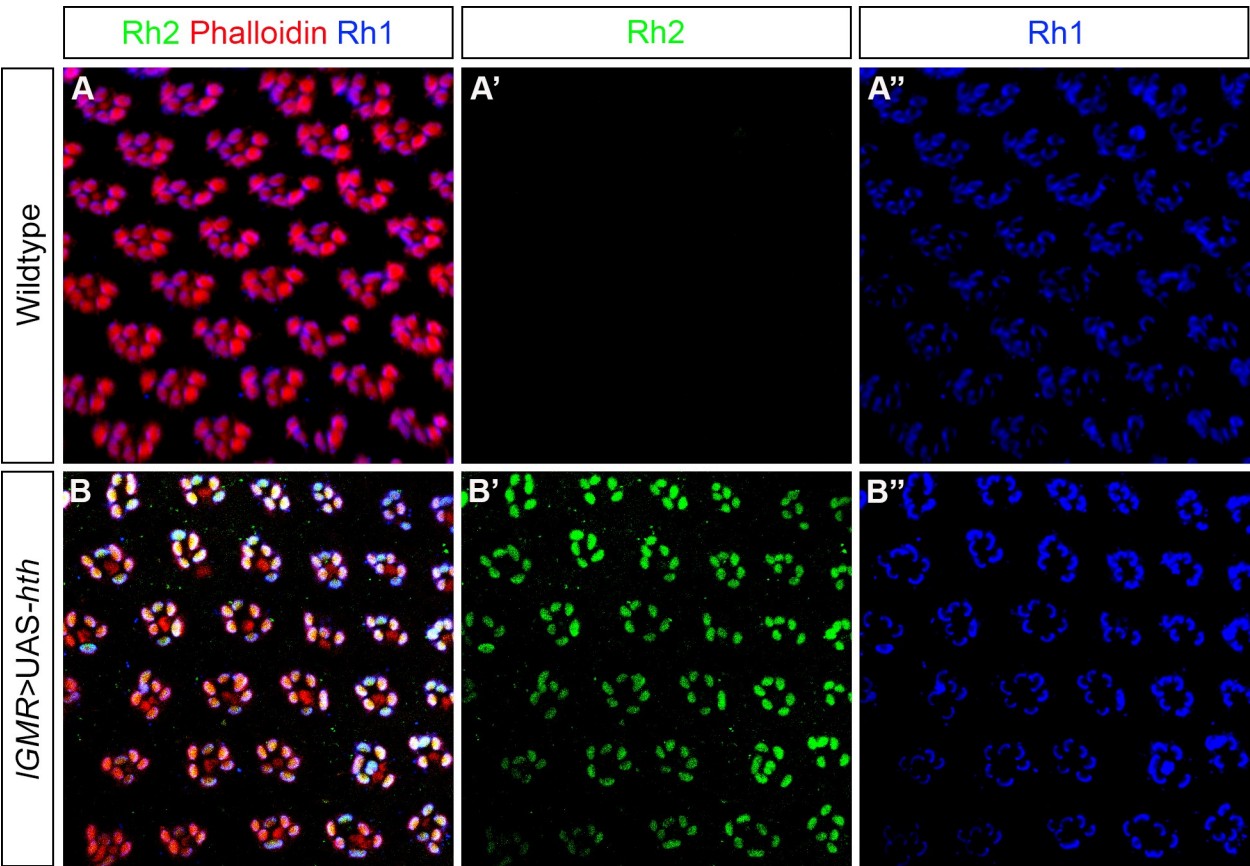

**Fig 4. Hth misexpression phenotype in the retina.** (A) Antibody staining to show expression of Rh2 and Rh1 in the wildtype retina. Rh2 (green) is normally not expressed in the retina (A') whereas Rh1 (blue) is expressed in the outer PRs of retina (A"). PRs of the retina were stained with phalloidin (red). (B) Expression of Rh1 and Rh2 in the *hth* overexpression retina by *lGMR*-Gal4. Rh2 expression (green) can now be ectopically seen in the outer PRs of retina (B') whereas Rh1 expression (blue) remained unaffected (B").

expressed in all outer PRs. However, we found that Rh1 expression was unaffected and still being expressed in all outer PRs (Fig 4B, 4B' and 4B"). We also found that ectopic Rh2 expression was only limited to the outer PRs (and not inner PRs) suggesting that only Rh1 expressing PRs were competent to induce Rh2 expression in the retina, while inner PRs were not.

To investigate if this process is cell autonomous, we generated outer PR versus inner PR Hth gain-of-function clones in the retina using the "flip-out" technique [43] (Materials and Methods for details). Flip-out clones were marked by presence of GFP expression. We found that when clones were generated in the inner PRs, Rh2 expression was not induced (Fig 5A, 5A' and 5A"). However, when clones were generated in the outer PRs, Rh2 expression was ectopically induced in all GFP expressing clones (Fig 5B, 5B' and 5B"). Thus, ectopic Rh2 expression in Rh1 expressing PRs of the retina is a cell-autonomous process. Expression of Hth in inner PR clones did not have any effect on Rh1 expression in the neighbouring outer PRs (Fig 5C, 5C' and 5C") and Rh1 expression was maintained when clones were generated in the outer PRs (Fig 5D, 5D' and 5D").

## Role of Scrib and Ets65A as potential repressors of Rh1 in the ocelli

To understand if Hth cooperates with any of its known interactors to regulate Rh1 and Rh2 expression in the ocelli, we performed a knockdown mini-screen of previously identified Hth

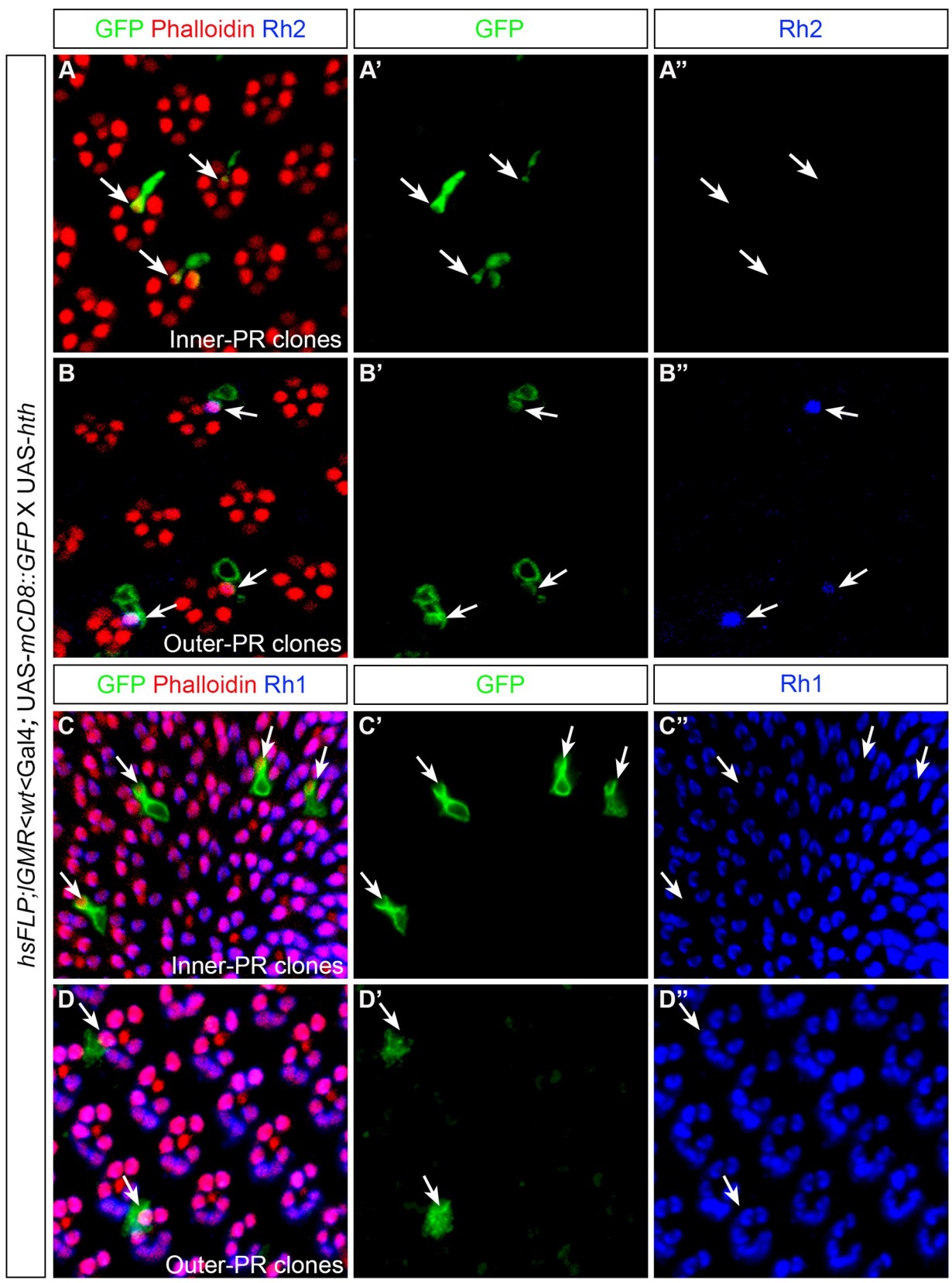

**Fig 5. Hth overexpression clones in the retina.** (A, B, C, D) Antibody staining to mark *hth* overexpression in clones (green) in the retina generated by the flip-out technique. (A, A') Antibody staining against GFP to identify inner PR clones of the retina, also co-stained by Rh2 (blue) and Phalloidin (red). Rh2 expression was not induced (A") when clones were generated in the inner PRs. (B, B') Antibody staining against GFP to identify outer PR clones of the retina, also co-stained by Rh2 (blue) and Phalloidin (red). When clones were generated in the outer PRs, Rh2 expression was induced (B"). White arrows marked inner and outer PR clones (A, A', B, B') and their corresponding Rh2 expression. (C, C', D, D') Antibody staining against GFP to identify inner and outer PR clones of the retina, also co-stained by Rh1 (blue) and Phalloidin (red). When clones were either induced in the inner PRs (C, C') or outer PRs (D, D') of the retina, Rh1 expression is unaffected (C", D").

interactors (mentioned in Flybase; S6 Fig) by *lGMR*-Gal4 and assayed for Rh1 and Rh2 expression in ocellar PRs. In agreement with previous findings, Rh2 expression was lost from the ocelli in *calmodulin-binding transcriptional activator (camta)*, *longitudenals lacking (lola)* and *defective proventriculus (dve)* knockdown [40]. However, we did not observe a gain of Rh1 in any of these knockdowns (Fig 6B, 6C and 6D). Surprisingly, in the mini-screen, we found that knockdown of the scaffolding protein Scribble (Scrib) [44,45] and the Ets domain transcription factor Ets65A [46,47] resulted in a gain of Rh1 expression in ocellar PRs without changing Rh2 expression (Fig 6E and 6F).

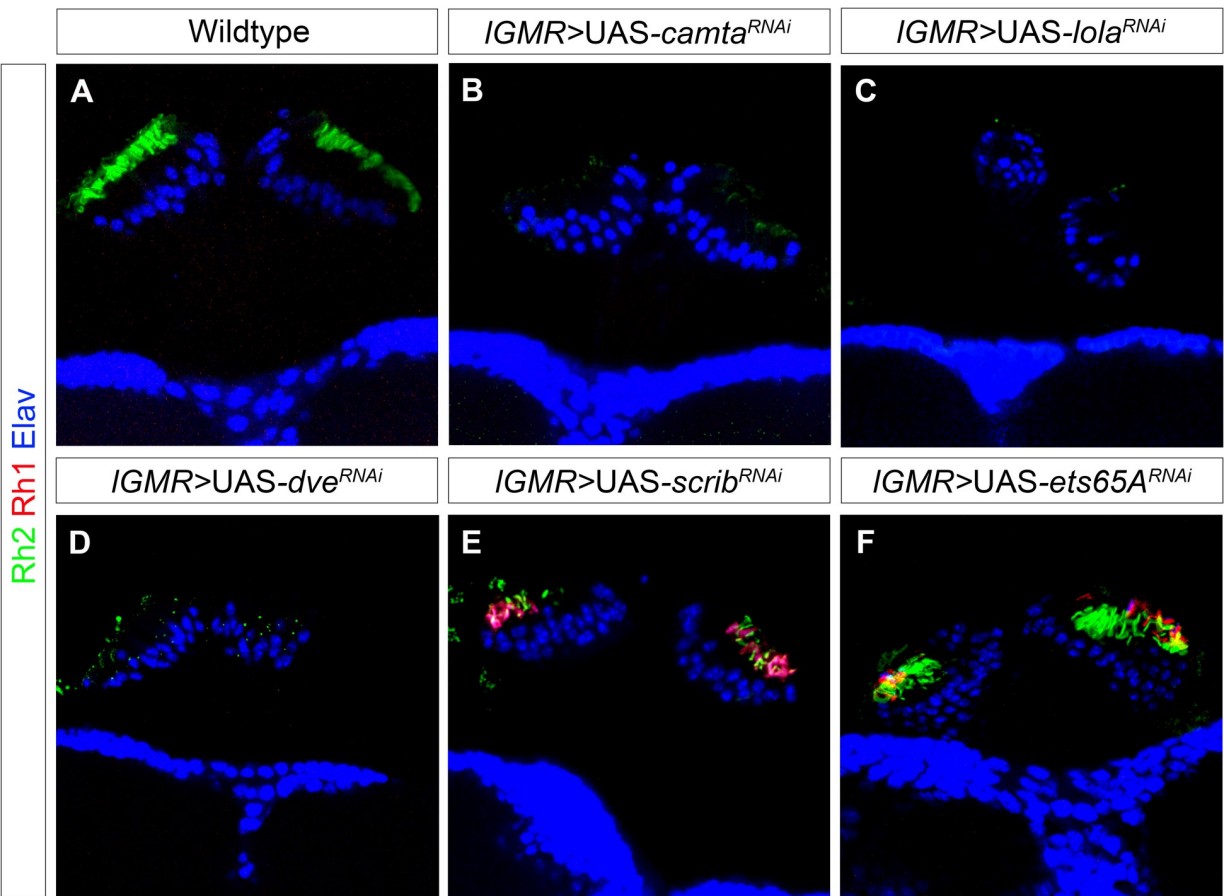

**Fig 6. Knockdown screening of Hth interactors in the ocelli.** (A) Expression of Rh1 (red) and Rh2 (green) in the wildtype ocelli. Ocellar PRs normally express Rh2 and Rh1 is usually absent. Knockdown of *camta* (B), *lola* (C) and *dve* (D) by *lGMR*-Gal4 showed loss of Rh2 expression (green) (previously found in [40]. However, Rh1 expression (red) was not ectopically induced in these knockdowns in the ocelli. Knockdown of *scrib* (E) and *ets65a* (F) showed a gain of Rh1 expression (red) in the ocelli whereas Rh2 expression (green) seemed to be unchanged in ocellar PRs.

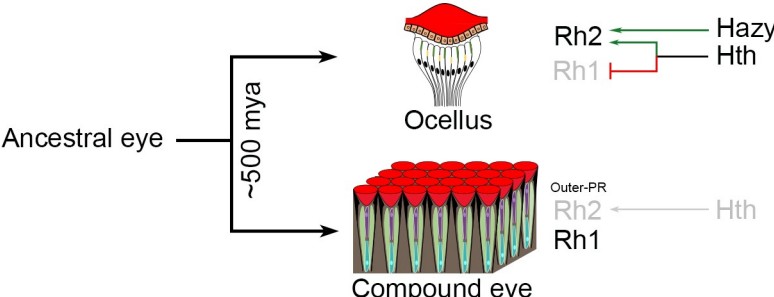

**Fig 7. Model for the binary Rhodopsin switch in the ocelli.** Schematic representation showing origin of ocelli and compound eye from an ancestral eye in insects around 500 million years ago. Rh2 is normally expressed in the ocelli and gets independently regulated by Hazy and Hth. Hth regulates the binary Rhodopsin switch to promote Rh2 expression in ocelli while repressing Rh1. Overexpression of *hth* is sufficient to promote Rh2 expression but only at the outer PRs of the retina.

Taken together, we have shown that Hth regulates a binary Rhodopsin switch in ocelli by promoting Rh2 expression at the cost of Rh1. We found that regulation of Rh1 and Rh2 is transcriptionally controlled by Hth acting together with its binding partner Exd and it is independent of Hazy (Fig 7). We also found that Hth regulates the Rhodopsin switch by operating through the promoters of *rh1* and *rh2*. We further demonstrated that misexpression of Hth in the retina modifies outer PRs to gain Rh2 expression resulting in co-expression of Rh1 and Rh2 in outer PRs and we show that this process is cell-autonomous. Finally, by knockdown mini-screen, we identified Scrib and Ets65A as potential repressors of Rh1 expression in the ocelli.

## Discussion

### Homothorax expression provides new insights to understand Rhodopsin fate in *Drosophila*

In insects, ocelli represent a fundamentally simpler visual organ whose spectral sensitivity is different from the compound eye. The unique spectral sensitivity of ocellar PRs in *Drosophila* is provided by the violet-sensing Rh2 [21]. The presence of this particular Rhodopsin exclusively in ocelli and not in the retina may explain why ocelli perform different functions than the compound eye. We have characterized the role of Hth during terminal differentiation of ocellar PRs and showed that Hth acts together with Exd and regulates a binary Rhodopsin switch in ocelli that promotes Rh2 expression and represses Rh1 expression. Hth is known to be expressed in the ocellar primordium of the early third instar larval (L3) eye antennal imaginal disc but gets downregulated later at mid- to late-L3 [48]. However, it is unknown how expression of Hth is re-induced in ocellar PRs. One possible hypothesis could be that a temporal change during metamorphosis such as a pulse of ecdysone hormone with additional signals induces Hth expression in ocelli. In our study, we show that Hth is expressed in all mature and terminally differentiated PRs of the ocelli.

Hth performs similar functions in PRs of the ocelli and the DRA of the retina: in ocelli, it induces Rh2 expression by repressing Rh1 whereas in the DRA of the retina it induces Rh3 by repressing inner PR Rhodopsins [25]. Loss of *hth* transforms the DRA into odd-coupled ommatidia where Rh3 is expressed in R7 and Rh6 in R8 suggesting that the Rh3/Rh6 pair represents the default state of Rhodopsins in inner PRs of the retina [25]. In ocelli, we find ectopic expression of Rh1 by loss of *hth* suggesting that Rh1 may be the default state in ocellar PRs.

Loss of Hth leads to ectopic expression of the R8 specific transcription factor Senseless in the DRA, which is otherwise not expressed in the wildtype DRA suggesting a fate change of the DRA to become odd-coupled ommatidia [34]. Interestingly, Hth loss in ocelli does not induce expression of outer PR-specific transcription factors (such as Svp and BarH1) suggesting that the ocellar PR fate may not have been changed.

## Genetic and molecular analysis of Hth in the ocelli during PR development

The DNA binding property of Hth is required for proper DRA fate in the retina [34]. Similarly, we observed that Hth regulates a binary Rhodopsin switch in ocelli by transcriptionally controlling *rh1* and *rh2* expression through their upstream DNA sequences. The homeodomain transcription factor Hazy controls Rhodopsin expression in the retina and in ocelli [40,41]. However, epistatic analysis showed that both Hth and Hazy act independently to control Rh2 expression in ocelli. Misexpression of *hth* in all developing PRs of the retina is sufficient to induce a fate switch where inner PRs are transformed into PRs of DRA [34]. Rh3 is expanded to all inner PRs whereas specific loss of Rh4, Rh5 and Rh6 was seen. However, outer PR fate was not changed suggesting that only those PRs, which were previously committed to become inner PRs during development, were responsive to Hth [34]. We additionally found that outer PRs, too were responsive to *hth* misexpression since they induce Rh2 expression while maintaining its default fate by expressing Rh1. However, the genetic program activated by Hth misexpression to induce Rh2 expression in outer PRs is still unknown. One hypothesis could be that Hth directly activates Rh2 expression in the ocelli but it requires an additional cofactor to repress Rh1 expression. This cofactor could be present in the ocelli but not in the retina and this would explain why knockdown of Hth activates Rh1 expression in the ocelli but is not sufficient alone to repress Rh1 expression in the retina. We additionally found that knockdown of Scrib and Ets65A induces Rh1 expression in the ocelli. However, further experiments will be required to establish the role of Scrib and Ets65A as potential repressors of Rh1 expression in ocelli versus retina.

## Rhodopsin duplication and role of Hth during ocellar diversification

Photoreception is achieved by expression of different opsins in conjunction with an array of proteins of the phototransduction cascade. Each PR contains a specific opsin, a light sensing protein that defines the particular wavelength of light to which a given PR will respond. In most insect species, opsins were mainly categorized based on their spectral sensitivity into: 1) UV-sensitive opsins 2) short-wavelength sensitive opsins and 3) long-wavelength sensitive opsins [49]. Presence of all three opsin types in most insect species may imply that the ancient retina was trichromatic [26]. Further phylogenetic analyses support ancestral trichromacy in insects and show that opsin diversity must have derived from a single ancestral opsin as a result of gene duplications [12]. In many insect species, spectral sensitivity of opsin gene clades is represented by more than one paralog such as the opsin clade consisting of three paralogs in *Drosophila* (Rh1, Rh2 and Rh6), five in the mosquito *Anopheles gambiae* and two in the honeybee *Apis mellifera* [12]. It is believed that a first tandem gene duplication in *Drosophila* separated Rh6 from its sister paralog Rh1/Rh2 and while Rh6 is retained in the inner PRs of compound eye, Rh1/Rh2 was associated with the outer PRs of retina and PRs of ocelli. A second gene duplication event led to the diversification of Rh1 and Rh2 by accumulation of further amino acid changes in their sequence. While Rh1 expression was retained in the outer PRs of the compound eye, Rh2 became associated with the PRs of the ocelli. Our results also suggest that the ability of Hth to bind to both promoters may have been a prerequisite during evolution to differentially regulate the expression of the two paralogs and thus, provide spectral

sensitivity of ocelli by promoting Rh2 expression and repressing Rh1 expression. However, it would be interesting to know if a change in the spectral sensitivity of ocelli upon Hth loss would in turn affect their specific functions. Also, if gene duplications of opsins occurred during evolution to allow the diversification of different spectral sensitivities in different visual organs, it would be interesting to know what kind of spectral sensitivity was present in the ancestral visual organ. We conclude that Hth may act as an evolutionary factor required in the ocelli to provide their unique spectral identity by expression of Rh2. To maintain this unique spectral identity, Hth controls a binary Rhodopsin switch to repress outer PR fate in ocelli by repressing Rh1 expression.

### The opposing regulatory function of Hth on its direct targets

We show here that Hth function both as a transcriptional activator of *rh2* and as a transcriptional repressor of *rh1*. Usually, Hth in conjunction with its binding partner Exd acts as a transcriptional activator. Interestingly, by knocking down *exd*, we also observed a loss of Rh2 and a gain of Rh1 expression suggesting that both factors act together. It was previously shown that binding of Hth to the DNA requires the presence of the co-factor Exd [50,51]. Therefore, likely regulation of Rh1 and Rh2 depend on the presence of both Hth and Exd binding to the regulatory region. Hth has also been shown to function in gene repression [52]. In this case the formation of the repressor complex occurs directly at the regulatory regions of the repressed gene and depends on the proximity of DNA-binding sites for different components of the complex in the regulatory region. The same mechanism could also explain the opposing regulatory effect that Hth has on *rh1* in comparison to *rh2* in the ocelli. Analysis of transcription profiles of different PR types would provide a list of Hth interactors expressed in the ocelli versus the outer PRs of the retina. Binding analysis of these interactors to the *rh1* and rh2 minimal promoter sequences would help to identify the potential repressor complex forming on the *rh1* promoter, helping to better understand the regulatory functions of Hth.

## Materials and methods

### Fly stocks

Wildtype Canton S flies have been used in this study. Other fly strains used were: UAS-*hth*[RNAi] (BL27655 and BL34637; we do see the Rhodopsin switch phenotype in both RNAi lines. However, all the experiments were done in BL27655), UAS-*exd*[RNAi] (BL29338 and BL34897; Rhodopsin switch phenotype was observed in both RNAi lines. However, all experiments were done in BL34897). For knockdown mini-screen, we used UAS-*RNAi* flies from Bloomington's *Drosophila* stock center and the stock numbers are mentioned in S6 Fig. Other fly strains are: UAS-*rh1* [37], UAS-*hth* [53], *rh1*(-252/+57)-*lacZ* [39], *rh2* (-309/+32)-*lacZ* [20], *rh2* (-293/+55)-*GFP* [40], *hazy*[-/-] [54], *hsFLP*; *lGMR*<wt<Gal4; UAS-mCD8::GFP (this "flip-out" Gal4 was a gift from Claude Desplan's lab). The following transgenic lines were made in this study: *rh2 (hth mut)-GFP*, *rh1-GFP* and *rh1 (hth mut)-GFP*. Flies were reared on standard food medium and at 25˚C. Knockdown flies were grown at 29˚C temperature. *hth* overexpression clones were generated by *FLP/FRT* system by using the "flip-out" technique [43]. A cross of *hsFLP*; *lGMR*<wt<Gal4; UAS-*mCD8::GFP with* UAS-*hth* was initiated and flies were grown at normal 25˚C. Temporal gene expression in the retina was initiated by heat shock at 37˚C for half an hour at the late pupal stage to induce expression of the Flip recombinase (*FLP*). FLP recognises FRT sites in the *lGMR*<wt<Gal4 cassette and it removes the cassette to activate Gal4 that induced overexpression of *hth*. After heat shock, vials were placed back at 25˚C and freshly hatched flies were dissected. Overexpression clones in the retina were marked by GFP expression.

## Generation of *rh1* and *rh2* reporter transgenes

The *rh1* minimal promoter (-247 to +73) was PCR amplified from genomic DNA with primers "rh1 enh Kpn fw" (gcggtacCTGGAGACTCAAGAATAATACTCGGCCAG) and "rh1 enh Xba re" (gatctagAGGGTTCCTGGATTCTGAATATTTCACTG) and cloned into pBluescript vector using the KpnI and XbaI sites added to the primers. For cloning of the *rh2* minimal promoter see [40]. The same *rh1* and *rh2* promoter fragments were *in vitro* synthesized (BioCat) altering the sequences of the potential Hth binding sites. The first Hth site in the *rh1* promoter (TGACAT) was changed to TaAgcT creating a HindIII restriction site. The second Hth site in the *rh1* promoter (CTGTCG) was changed to CTaaaG. The first Hth site in the *rh2* promoter (GGACAG) was changed to GtttAG, the second Hth site in the *rh2* promoter (GTGTCA) was changed into agcTgA and the third Hth site (CTGTCC) was changed to CTaaaC. Both versions of the two enhancers were cloned into a GFP reporter plasmid containing *eGFP*, a *mini-white* marker *and an attB* site kindly provided by Jens Rister. The plasmids were injected into nos-φC31; attP40 flies for integration on the second choromosome using the φC31 site-specific integration system [55].

## Immunohistochemistry

Adult ocelli were dissected and stained by a protocol published in [40] whereas dissection and immunohistochemistry of adult retinas were done according to [56]. After the immunostainings, tissue samples were mounted by using Vectashield H-1000 (Vector laboratories). Primary antibodies and their dilutions were as follows: Rabbit anti-Rh2 1:100 [40], Rabbit anti-Rh6 1:10,000 [57], Rabbit anti-Hth 1:500 [58], Chicken anti-βGal 1:1000 (Abcam), Chicken anti-GFP 1:2000 (Life technologies), Rabbit anti-Hazy 1:500 [54], Mouse anti-Svp 1:100 [59], Rat anti-BarH1 1:200 [60], Mouse anti-Rh1 1:20, Rat anti-Elav 1:20 and Mouse anti-Chp 1:20 (Developmental Studies Hybridoma bank). Phalloidin conjugated with Alexa-568 and Alexa-647 (Sigma-Aldrich, Life Technologies) marks F-actin and were used (1:5000) during incubation with secondary antibodies. The following secondary antibodies were used: Goat anti-rabbit, Goat anti-mouse, Goat anti-rat and Goat anti-chicken that are conjugated with Alexa-488, Alexa-555 and Alexa-647 (Jackson Immunoresearch). All secondary antibodies are used at 1:200 dilution.

## Confocal microscopy and Image analysis

Tissue samples were imaged with a Leica TCS SP5 confocal microscope at a resolution of 1024x1024 pixels and optical sections were taken in the range of 1–2 μm depending on the sample size. Images were further processed and analysed in Fiji/ImageJ and Adobe photoshop 2020 software.

## Supporting information

**S1 Fig. Phylogenetic analysis of opsin (*rh1*, *rh2* and *rh6*) duplications in dipterans.** Phylogenetic tree by maximum likelihood method showing *rh1*, *rh2* and *rh6* duplications in different dipteran species, such as *Drosophila melanogaster*, *Ceratitis capitata*, *Musca domestica*, *Glossina palpalis*, *Lucilia cuprina*, *Aedes aegypti* and *Anopheles gambiae*. Opsin paralog groups were named according to *Drosophila* nomenclature. Amino acid sequences of Rhodopsins were used for making the phylogenetic tree and these sequences were available either in Feuda lab on bitbucket (https://bitbucket.org/Feuda-lab/opsin_diptera/src/master/) [33] or in NCBI (National Center for Biotechnology Information). The Phylogenetic tree was made (by using

MUSCLE online tool phylogeny.fr) and Rh5 sequence from *Drosophila* was taken as outgroup.
(TIF)

**S2 Fig. Independent roles of Hazy and Hth to regulate Rh2 expression in ocelli.** (A) Overexpression of *rh1* in all PRs of ocelli by using pan-photoreceptor *lGMR*-Gal4. Antibody staining against Rh2 (green) and Rh1 (red) was performed to show that overexpression of *rh1* does not inhibit Rh2 expression in ocelli. (B) Antibody staining to show expression of Rh2 (green) and Rh1 (red) in *hazy*$^{-/-}$ null mutant ocelli. Rh2 expression is lost in *hazy*$^{-/-}$ mutants but they don't show ectopic Rh1 expression. (C) Antibody staining to show Hth expression (blue) in *hazy*$^{-/-}$ mutant ocelli, also marked by Chp (green). Hth expression is not affected in ocellar PRs in *hazy*$^{-/-}$ mutants. (D) Antibody staining to show Hazy expression (blue) in *hth* knockdown ocelli. Ocelli in *hth* knockdown are marked by staining against Rh1 (red). Hazy is expressed in *hth* knockdown ocelli that have lost Rh2 and gained Rh1 expression.
(TIF)

**S3 Fig. Knockdown of Hth does not alter ocellar PR cell fate.** (A, B) Antibody staining against Svp (green) in the wildtype and in *hth* knockdown ocelli. Svp is normally expressed in R3/R4 and R1/R6 pairs of the developing retinal PRs [39] and not in wildtype ocelli (A). During the Rhodopsin switch by *hth* knockdown in ocelli, Svp is still not expressed in ocellar PRs (B). (C, D) Antibody staining against BarH1 (green) in the wildtype and in *hth* knockdown ocelli. BarH1 is normally expressed in R1/R6 pair of the developing retina [38] and not in wildtype ocelli (C). Upon Rhodopsin switch by *hth* knockdown, BarH1 is still not expressed in the ocellar PRs (D). Ocellar PRs are marked by antibody staining against Elav (blue, in A and B) and Chp (red, in C and D).
(TIF)

**S4 Fig. Alignment of the *Rhodopsin 1* promoters of 12 *Drosophila* species.** Conserved residues are depicted on a grey background. For the reporter constructs we cloned a 320 bp fragment ranging from nucleotides 2 to 321 of the *D. melanogaster* sequence in front of GFP. The two potential Hth binding sites are outlined in red. The RCSI is outlined in blue. The first Rh1 exon of *D. melanogaster* is outlined in black. In all twelve species the translation start (Met, arrow) is directly followed by an intron. Species used: *D. melanogaster (Dmel), D. simulans (Dsim), D. sechellia (Dsec), D. yakuba (Dyak), D. erecta (Dere), D. ananassae (Dana), D. persimilis (Dper), D. pseudoobscura (Dpse), D. wilistoni (Dwil), D. grimshawi (Dgri), D. mojavensis (Dmoj), D. virilis (Dvir)*.
(TIF)

**S5 Fig. Alignment of the *Rhodopsin 2* promoters of 12 *Drosophila* species.** Conserved residues are depicted on a grey background. For the reporter constructs we cloned the 348 bp fragment of *D. melanogaster* from the endogenous Sal I restriction site (underlined in green) to the last nucleotide before the translation Start (Met, arrow) in front of GFP. This promoter sequence partially overlaps the last exon of the neighbouring gene *CG14297*. The three potential Hth binding sites are outlined in red. The first site is located within the coding sequence of *CG14297*. The second site which is only conserved within the melanogaster group (top five species) is located at the stop codon (asterisk) of *CG14297*, and the last site is located at the end of the 3'UTR. The RSCI (outlined in blue) is located within the 80 bp sequence between the end of the *CG14297* 3'UTR and the transcription start of *Rhosopsin 2*. The exons of *D. melanogaster* are outlined in black. Species used: *D. melanogaster (Dmel), D. simulans (Dsim), D. sechellia (Dsec), D. yakuba (Dyak), D. erecta (Dere), D. ananassae (Dana), D. pseudoobscura (Dpse), D. persimilis (Dper), D. wilistoni (Dwil), D. mojavensis (Dmoj), D. grimshawi (Dgri), D.*

*virilis (Dvir).*
(TIF)

**S6 Fig. A table for knockdown mini-screening candidates in the ocelli.** The table consists of candidates either known to have interactions with Hth (information from Flybase) or they have been previously found to regulate Rh2 expression [40]. For knockdown screening, flies were ordered from Bloomington's *Drosophila* stock center and the stock numbers are listed in the table.
(TIF)

## Acknowledgments

We thank C. Desplan, J. Rister, A. Salzberg, T. Cook, D. Vasiliauskas, Developmental Studies Hybridoma Bank (DSHB) and Bloomington Stock Center for providing flies and antibodies. We thank UniFr bioimage team for maintaining imaging facility and all members of Egger and Sprecher lab for fruitful discussions.

## Author Contributions

**Conceptualization:** Roumen Voutev, Richard S. Mann, Simon G. Sprecher.

**Data curation:** Abhishek Kumar Mishra.

**Formal analysis:** Cornelia Fritsch, Roumen Voutev, Richard S. Mann.

**Funding acquisition:** Richard S. Mann, Simon G. Sprecher.

**Investigation:** Abhishek Kumar Mishra.

**Methodology:** Abhishek Kumar Mishra.

**Project administration:** Simon G. Sprecher.

**Supervision:** Simon G. Sprecher.

**Visualization:** Abhishek Kumar Mishra, Cornelia Fritsch.

**Writing – original draft:** Abhishek Kumar Mishra, Cornelia Fritsch, Simon G. Sprecher.

**Writing – review & editing:** Abhishek Kumar Mishra, Cornelia Fritsch, Simon G. Sprecher.

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
