## [Decision Letter · Decision Letter 0]

22 Mar 2021

Dear Dr Sprecher,

Thank you very much for submitting your Research Article entitled 'Homothorax Controls a Binary Rhodopsin Switch in Drosophila Ocelli' to PLOS Genetics.

The manuscript was fully evaluated at the editorial level and by independent peer reviewers. The reviewers appreciated the attention to an important problem, but raised some substantial concerns about the current manuscript. Based on the reviews, we will not be able to accept this version of the manuscript, but we would be willing to review a much-revised version. We cannot, of course, promise publication at that time.

If you decide to revise the manuscript for further consideration at PLOS Genetics, please aim to resubmit within the next 60 days, unless it will take extra time to address the concerns of the reviewers, in which case we would appreciate an expected resubmission date by email to plosgenetics@plos.org.

[LINK]

We are sorry that we cannot be more positive about your manuscript at this stage. Please do not hesitate to contact us if you have any concerns or questions.

Yours sincerely,

Fengwei Yu

Associate Editor

PLOS Genetics

Gregory P. Copenhaver

Editor-in-Chief

PLOS Genetics

While the manuscript very nicely shows that the homeodomain protein Hth differentially regulates the expression of rh1 and rh2 in the ocelli as a potential binary switch, the reviewer #2 and #3 also pointed out that understanding of how Hth differentially regulates rh1 and rh2 expression would significantly strengthen the conclusion and provide mechanistic insight. The editors agree with the reviewers on their suggestion. We thus suggest the authors to conduct some additional experiments to explore the potential mechanisms underlying the opposite transcriptional regulation of rh1 and rh2. As Hth is part of a transcriptional repressor complex En-Hth-Exd, the author may consider examining the involvement of Exd, En and other potential Hth interactors. A mini-RNAi screen was suggested by the reviewer #2. Moreover, as the reviewer #3 suggested, the authors should also examine any potential binding sites for this repressor protein in rh1 and rh2 loci.

Reviewer's Responses to Questions

**Comments to the Authors:**

Reviewer #1: Evolutionary origin of the animal visual system is an important problem in biology. Evolutionary diversification of compound eye and ocelli of insects is a key process to solve this issue. In this study, the authors focused on the gene regulatory mechanism that specify the ocelli specific Rhodopsin2 (Rh2) expression and the compound eye specific Rh1 expression. They demonstrate that an evolutionarily conserved Homothorax (Hth) activates Rh2 transcription and represses Rh1in ocelli through the Hth-binding sites in the Rh1/2-promoters. The question is of general interest and the results are clearly demonstrated by a series of genetic experiments. However, the authors need to address the following concerns prior to the publication of the manuscript.

1. In Introduction, Results and Discussion sections, they indicate that Rh2 is UV-blue sensitive (418-506nm). According to the definition in line 111-113, it should be regarded as Blue-sensitive (or middle-wavelength according to line 294-297). However, in line 119-121, they indicate that Rh2 is long-wavelength (LW)-opsin (In Drosophila, PRs in the ocelli express the LW-opsin Rh2 that is sensitive to a wavelength range of 418-506 nm). I think the authors mean that Rh2 belongs to the LW-opsin group in the phylogenetic tree. This is confusing because the group of Rh1/2/6 contains rhodopsins of diverse wavelength. The groups in the phylogenetic tree should be named differently to avoid possible confusion.

2. The finding that hth activates Rh2 transcription and represses Rh1 in ocelli while it only activates Rh2 in compound eye is interesting. According to the results of Rh1 promoter-GFP and Rh2 promoter-GFP experiments, Hth activates Rh2 transcription and represses Rh1through the Hth binding sites. The authors should discuss why Hth does not repress Rh1 transcription in the compound eye.

3. In Figs 1C-E, 2A-B and the other similar figures, the authors should explain why Rhodopsin expression does not overlap with nuclear signals such as Elav.

4. Line 204, 'mutating the’.

Reviewer #2: The manuscript by Mishra and colleagues addresses the interesting questions of how rhodopsin genes are restricted to individual visual organs or classes of photoreceptors. The authors focus on the regulation of the Rh1 and Rh2 rhodopsin genes in the fruit fly Drosophila melanogaster. These two rhodopsin genes arose from a duplication event with Rh1 being expressed in the compound eye and Rh2 being expressed in the ocelli. The authors show convincingly that the homeodomain transcription factor Homothorax (Hth) is expressed in the ocelli where it activates expression of Rh2 and suppresses expression of Rh1. They go on to show that forced expression of Hth in the compound eye activates Rh2. Together the data suggest that Hth regulates a binary switch that maintains Rh2 expression in the ocellar photoreceptors.

The paper was a pleasure to read and I have only a couple of small suggestions for the authors.

1. Since Hth is known to interact with Extradenticle it might be interested if the authors could remove Exd using RNAi and see if Rh2 expression is turned off in the ocelli.

2. The authors note that Hth physically interacts with Engrailed to repress target gene expression. I think the authors should downregulated En with RNAi and see if Rh1 expression is activated in the ocelli.

3. The authors could also do a quick mini-screen of factors that are known to physically interact with Hth – I believe that information is listed within Flybase. The authors could use RNAi lines (where available) to knock them down with lGMR-GAL4 and assay for Rh1 and Rh2 expression.

Otherwise, the paper is really well written and the data within the figures are beautiful.

Reviewer #3: In Drosophila, the rhodopsin Rh1 is expressed in the outer PR in the retina, while Rh2 is expressed in the ocelli. In this manuscript, the authors showed that the homeodomain protein Hth is expressed in the ocelli and induces rh2 transcription and suppresses rh1 transcription. The major results are (1) knockdown of Hth induced Rh1 (and rh1-lacZ) and eliminated Rh2 (and rh2-lacZ) in ocelli, (2) mutating the putative Hth-binding sites in the rh1 and rh2 minimal promoters eliminated rh2(hth_mut)-GFP expression and induced rh1(hth_mut)-GFP expression in ocelli. The effect of Hth is not through change in cell identity and not through regulation between rh1 and rh2. Therefore, it is concluded that hth controls a binary rhodopsin switch in ocelli.

In the retina, hth is not expressed. Ectopic expression of hth cell-autonomously induced Rh2 expression only in the outer PRs. However, Rh1 was not suppressed. Therefore, the “binary switch” does not operate in the retina.

The data are clean and nice. The results are clear cut. The finding that the choice of Rh1 and Rh2 expression in ocelli depends on Hth is nice but not so novel as Hth is known to regulate rh3 expression in the inner PRs in the dorsal rim area (DRA) of the compound eye. The novelty is only Hth exerts opposite transcriptional effect on the two genes. But this again is not novel if the two genes were not functionally related. The regulation by Hth on the two rhodopsin gens seems independent of each other. The evolutionary implication that Hth is involved in rhodopsin diversification needs more evidence.

Comments;

1. For the phylogenetic analysis of rhodopsin genes (Fig. 1B), the authors should expand their analysis to include rhodopsin genes from different representative branches of arthropods, perhaps even beyond the arthropods. Many genome sequences are now available.

2. The conclusion that Hth directly regulates rh1 and rh2 transcription is based on mutating the putative Hth binding sites in these promoters. The authors need to add experiments to show the direct binding of Hth to these binding sites.

3. The flp-out-clones with Hth expression were analyzed in the retina. Similarly, hth LOF clones should be performed in the ocelli to test whether the effects in ocelli on rh1 and rh2 expression is also cell-autonomous.

4. Based on the authors’ findings, Hth directly induces rh2 transcription but suppresses rh1 transcription. How to explain the opposite transcriptional regulation? Hth has been shown to be a transcriptional activator (Inbal et al., 2001). It can also participate in a transcriptional repressor complex (e.g. En-Hth-Exd; Kobayashi et al., 2003; Fujioka et al., 2012). The authors made some discussions, but did not pursue an answer. Since the rh1 and rh2 minimal promoter fragments can reflect the endogenous regulation, their sequence should be analyzed. Does the rh1 promoter also contains binding sites for repressor protein? Such binding site may be absent in the Rh2 promoter.

5. In ocelli, Hth directly represses Rh1 and activates Rh2 transcriptionally. However, in the retina outer PRs, ectopic Hth can induce Rh2 expression but not repress Rh1 expression. This may suggest that the transcriptional repression of Rh1 requires some cofactor, e.g. a transcriptional repressor that can interact with Hth. This may further suggest such transcriptional repressor is present in the ocelli but absent in the outer PR of the retina. Can the authors provide some insight in this issue? For example, is the putative repressor (based on the binding sites in Rh1 promoter) expressed in the ocelli but not in retina?

6. In addition to the phylogenetic analysis of the rhodopsin coding region (not clear whether only the ORF were compared), the rh1 and rh2 minimal promoter sequence should also be analyzed phylogenetically in additional insect (and arthropods, if possible).

7. For readers not familiar with the fly visual system, it would be nice if the authors can use an image to show the orientation and mark the different parts of the visual system, so the readers can understand what to look at in the figures.

8. The Rh1 results may also be interpreted as rh1 being a default expression for all PRs, but is suppressed by Hth in the ocelli and perhaps suppressed by some unknown factor in the inner PRs.

9. I have some reservation on the use of “binary switch”, but I would not make a strong point.

**Have all data underlying the figures and results presented in the manuscript been provided?**

Reviewer #1: Yes

Reviewer #2: Yes

Reviewer #3: Yes

PLOS authors have the option to publish the peer review history of their article (what does this mean?). If published, this will include your full peer review and any attached files.

Reviewer #1: **Yes: **Makoto Sato

Reviewer #2: No

Reviewer #3: No

---

## [Decision Letter · Decision Letter 1]

2 Jun 2021

Dear Dr Sprecher,

Thank you very much for submitting your Research Article entitled 'Homothorax Controls a Binary Rhodopsin Switch in Drosophila Ocelli' to PLOS Genetics.

The manuscript was fully evaluated at the editorial level and by independent peer reviewers. The reviewers appreciated the attention to an important topic but identified some concerns that we ask you address in a revised manuscript

We therefore ask you to modify the manuscript according to the review recommendations. Your revisions should address the specific points made by each reviewer.

[LINK]

Yours sincerely,

Fengwei Yu

Associate Editor

PLOS Genetics

Gregory Copenhaver

Editor-in-Chief

PLOS Genetics

Reviewer's Responses to Questions

**Comments to the Authors:**

Reviewer #1: The paper was properly revised.

However, I still have a concern about the description of 'long wavelength-opsin'. I don't get what was changed in the revised result and discussion sections. If 'long wavelength-opsin' is preferred for the name the group of rhodopsins Rh1/2/6 due to the historical reason of this field, it should be mentioned in the text.

Reviewer #2: The authors have addressed all of my comments. I recommend publication in PLOS Genetics. I am looking forward to seeing the paper in print!

Reviewer #3: The revision has addressed all of my comments.

**Have all data underlying the figures and results presented in the manuscript been provided?**

Reviewer #1: Yes

Reviewer #2: Yes

Reviewer #3: Yes

PLOS authors have the option to publish the peer review history of their article (what does this mean?). If published, this will include your full peer review and any attached files.

Reviewer #1: No

Reviewer #2: No

Reviewer #3: No

---

## [Editor Report · Decision Letter 2]

2 Jul 2021

Dear Dr Sprecher,

We are pleased to inform you that your manuscript entitled "Homothorax Controls a Binary Rhodopsin Switch in Drosophila Ocelli" has been editorially accepted for publication in PLOS Genetics. Congratulations!

Yours sincerely,

Fengwei Yu

Associate Editor

PLOS Genetics

Gregory P. Copenhaver

Editor-in-Chief

PLOS Genetics

Comments from the reviewers (if applicable):

**Data Deposition**

http://datadryad.org/submit?journalID=pgenetics&manu=PGENETICS-D-21-00286R2

**Press Queries**

---

## [Editor Report · Acceptance letter]

22 Jul 2021

PGENETICS-D-21-00286R2 

Homothorax Controls a Binary Rhodopsin Switch in Drosophila Ocelli 

Dear Dr Sprecher, 

We are pleased to inform you that your manuscript entitled "Homothorax Controls a Binary Rhodopsin Switch in Drosophila Ocelli" has been formally accepted for publication in PLOS Genetics! Your manuscript is now with our production department and you will be notified of the publication date in due course.

With kind regards,

Andrea Szabo

PLOS Genetics

On behalf of:
